# Insect Meal as an Alternative to Protein Concentrates in Poultry Nutrition with Future Perspectives (An Updated Review)

**Qurat Ul Ain Sajid** [1,*] , **Muhammad Umair Asghar** [2,*] , **Haneef Tariq** [1] , **Martyna Wilk** [2] and **Arkadiusz Płatek** [2]

1   Department of Plant Production and Technologies, Niğde Ömer Halisdemir University, 51240 Niğde, Turkey; haneeftariq1@gmail.com
2   Department of Animal Nutrition and Feed Science, Wrocław University of Environmental and Life Sciences, 51-630 Wrocław, Poland; martyna.wilk@upwr.edu.pl (M.W.); 121483@student.upwr.edu.pl (A.P.)
*   Correspondence: qiratsajid50@gmail.com (Q.U.A.S.); muhammad.asghar@upwr.edu.pl (M.U.A.); Tel.: +48-786-650-472 (M.U.A.)

**Abstract:** In recent years, interest has grown among poultry nutritionists in using alternative protein sources, such as insect meal, to meet the protein requirements of poultry due to sustainability concerns surrounding traditional protein sources such as soybean and fish meal. Insect meal can be produced from different insects, including black soldier fly, mealworms, and house crickets, and its nutrient composition varies depending on the insect species, the substrate they are reared on, and the production method. This review article provides an updated overview of insect meal as a new form of protein concentrate in poultry diets, including its nutritional value, advantages, challenges, and future prospects. Insect meal has been shown to be a rich source of protein, amino acids (lysine, methionine), and minerals (calcium, phosphorus, zinc), with a high digestibility rate, making it a valuable feed ingredient for poultry production. Additionally, using insect meal in poultry feed could reduce the cost of production and the environmental impact on the industry. Furthermore, the use of insect meal has the potential to improve the growth performance and meat quality of poultry species. However, several challenges related to large-scale insect production, legal regulatory frameworks, and consumer acceptance need to be addressed. Future research and development could help overcome these challenges and increase the adoption of insects as a potential source of protein in poultry feed. This review provides an updated and comprehensive overview of insects as a potential source of protein for poultry nutrition and highlights the possible perspectives of insect meal to contribute to a more sustainable and efficient poultry production system. While challenges remain, the utilization of insect meal in poultry feed has the capability to enhance the sustainability and efficiency in the poultry industry. Hence, insect meal emerges as a highly encouraging protein alternative, offering sustainable prospects for its utilization within the poultry sector. However, advancements in insect production technology and efficiency have the potential to raise the production scale while lowering prices, making insect meals more affordable compared to conventional protein sources. Based on the comprehensive analysis, it is recommended to further explore the practical implementation of insect meal as a reliable and efficient means of supplying protein in poultry nutrition.

**Keywords:** insects meal; poultry nutrition; protein alternatives; sustainability; digestibility; future challenges

## 1. Introduction

Over the last few decades, the human population has grown incessantly, and it is estimated that by the end of this century the population will reach 12.3 billion [1,2]. According to Roser et al. [1], the global population will approach nearly 10 billion by the year 2050, highlighting the crucial role that the food production industry, especially animal protein production, will play in ensuring food and nutrition security. In order to fulfill the food

demands of such a large population, a substantial quantity of high-quality food is needed. Poultry eggs and meat can fulfill the protein needs of people and are affordable food sources with a short-term production cycle. Therefore, the future will face a continued demand for poultry feed. Poultry meat is the most economical and consumable source of animal protein for many communities globally, as noted by Asghar et al. [3]. The feed market in the Asia-Pacific region is projected to reach140 billion USD by 2024 [4]. Nevertheless, the escalating costs of poultry production, caused by the increasing expense of feed, pose a risk to the sector, as observed by Naımatı et al. [5]. However, as the demand for products derived from poultry grows, feed resources may become limited to sustain such growth in poultry production and operations, as observed by Thirumalaisamy et al. [6].

Soybean and fish meals, which are conventional protein sources, have been associated with notable environmental concerns such as greenhouse gas emissions, deforestation, and water pollution. As a result, an increasing interest has emerged in using alternative protein sources, including plants and animal-based proteins (insect meal) [7,8]. Plant-based proteins are becoming increasingly popular as an eco-friendly substitute for traditional sources of protein for livestock. Soybean-based meal is one example, used as a common source of protein in livestock feed, accounting for around 65% of the global proportion of protein meal production attributed to soybean meal based on the available data and industry reports [9]. Moreover, researchers have been exploring alternative plant-derived protein sources, such as canola, peas, and algae. Pea protein, for example, is abundant in indispensable amino acids with a lower environmental impact than soybean production [10,11].

The market for protein ingredients has grown significantly in recent years, with a projected value of 38 billion USD in 2019. It is projected to experience a growth rate of 9.1% from 2020 to 2027 [12]. Animal protein consumption has significantly increased in recent years, driven by factors such as population growth, urbanization, and rising incomes in developing nations [13]. Based on a study published by the United Nation Food and Agriculture Organization, the global meat demand is projected to increase by 1.3% per year over the next decade [9]. Protein is an essential nutrient for poultry, and traditional protein sources such as soybean and fish meal have several disadvantages, including the high cost of fish meal, limited availability, and long term availability concerns. The utilization of soybean meal (SBM) in poultry rations is prevalent, primarily attributed to its improved crude protein content and amino acid profile [14]. Protein is the second most important component in the diet of poultry. In light of the price fluctuations of SBM and the continuous rise in feed costs, researchers have been investigating alternative sources of protein. The substitution of SBM with alternative sources of protein in poultry meals has the potential to alleviate the competition between humans and livestock for soybean resources and to enhance the output of animal protein [15]. Legumes, cereals, and seeds are essential plant protein sources for humans, especially in a vegetarian diet. Meat and bone meal is often used in poultry diets; however, it is less digestible than soybean meal. In light of the particular dietary requirements of poultry, the digestibility of protein sources is an important element to consider [16]. Therefore, we need to find ways to increase insect-based protein diets without causing harmful effects on the environment. To utilize feed additives in chicken feed without negatively impacting the environment, it is critical to carefully choose and control the chemicals and implement advanced farming strategies to monitor and comply with environmental rules. Numerous ranges of feed additives that possess favorable nutritional benefits for poultry have been previously documented in the literature [17,18].

In recent times, insect meal as a source of protein in poultry diets is gaining attention, which has the potential to lower feed costs and boost profitability via the utilization of fresh insects for small-scale poultry production [19,20]. Insects offer a promising and subsistence solution for addressing the challenges associated with conventional protein sources, while providing comparable nutritional value and a reduced environmental footprint [21,22]. The contribution of poultry to the economies of many developing countries is significant and supports the livelihoods of many individuals, especially in rural areas [23]. Consuming

poultry products, including meat and eggs, is a significant means for humans to acquire animal protein [24].

Insect meal is another alternative protein source that has attracted interest nowadays. It has been found that insect meal is a reliable source of protein for animal and poultry feed, with research indicating that it can improve animal growth and reduce feed conversion ratios [25]. It is believed that insect meal offers a potential alternative protein source in feed. Moreover, there is a need to overcome some obstacles, such as regulatory constraints and consumer acceptability. However, the utilization of insects in livestock feed is already permitted in some countries, including the European Union and Canada [26].

The usage of poultry meat, particularly chicken, is predicted to keep rising in the next ten years, as it is currently the most popular white meat [3,27]. Poultry production is considered to have a relatively low environmental impact in comparison to other meats because it does not produce enteric fermentation and has a lower feed conversion ratio (FCR) [28]. While there are benefits to raising poultry, such as their efficient use of resources and faster growth rate, the costs of production are increasing. The high costs of poultry production can be attributed to various factors, with the cost of feed being one of the primary factors that accounts for a considerable portion of the overall production cost [29]. The primary reason for the high cost of poultry feed is the production and processing cost, market demand, and supply of soyabean globally [30]

Furthermore, the competition for limited resources among the food, feed, and fuel sectors, along with the impact of changing climatic factors, has significantly influenced the access to conventional feed ingredients such as soybean, fish meal, and cereals. This has resulted in a decrease in accessibility and a high level of volatility in feed resource prices, as highlighted by Mugwanya et al. [31]. In the last ten years, the lack of traditional feed resources has caused an increase in feed prices, making it necessary to search for alternative protein sources for poultry. The usage of insects as a source of feed for farmed animals holds great promise due to the nutritional benefits (essential amino acid profiles and high protein contents) of insects and possible environmental advantages of this farming practice, which mainly depends upon the substrate used for insect rearing and its impact on the quality of the insect meal, according to van Huis et al. [25]. However, Western customers do not generally agree regarding the consumption of insects. It is possible to collect insects in the wild or rear them commercially at a low cost, with a shorter production cycle, as noted by Oonincx et al. [32]. Although the natural harvesting of insects may be practical for small-scale or subsistence farming, commercial-scale insect production at a constantly low cost is still under consideration. The consumption of insects in human food has been prevalent in tropical countries for centuries, as stated by DeFoliart et al. [33]. In addition to serving as a source of protein, the fat collected from insects in the meal process of extraction can be utilized in the diets of chicken, reducing the need for soybean and various vegetable oils, according to [4,34]. As a result, the recent research has aimed to assess possible alternatives, including insects [35–37], bacteria [35,38], and organic by-products [39,40]. Insects gained the most attention among them because of their widespread usage in poultry feed and their easy production [41]. Common house flies, black soldier flies, yellow mealworms, and blowflies are among the insects that are the potential alternatives for protein sources in poultry diets, as highlighted by Čičková et al. [42]. Insects have high mineral contents, including zinc (Zn) and iron (Fe), as noted by Finke et al. [43], and contain an array of vitamins, including riboflavin, folic acid, cyanocobalamin, thiamine, and retinol traces, as described by Rumpold et al. [44]. Insects additionally possess a particular kind of peptide that shows an antioxidant action, which can be beneficial for the health of livestock, according to Schiavone et al. [45]. Water-soluble products based on insects exhibit exceptional antioxidant properties and have free radical neutralization characteristics, setting them apart from various plant- and animal-derived protein hydrolysate products, as observed by Di Mattia et al. [46].

This review provides an up-to-date overview of insect-based diets as novel protein concentrates in poultry nutrition, encompassing their nutritional value, benefits, obstacles,

and future prospects. This review strives to contribute significantly to the utilization of insect-based feeds in poultry farming, aiming to make a valuable contribution to the ongoing research on alternative protein sources. Therefore, several insect species have been intensively researched for the growth and development of insect-based feeds [47]. These species include the common housefly (*Musca domestica*), mealworm (*Tenebrio molitor*), black soldier fly (*Hermetia illucens*), locusts (*Schistocerca gregaria*, *Locusta migratoria*, and *Oxya* species), silkworm (*Bombyx mori*), mopane worm (*Gonimbrasia belina*), field cricket (*Gryllus bimaculatus*), Westwood insect (*Cirina forda*), grasshoppers (*Caelifera* (Suborder)), and earth worm (*Lumbricus terrestris*).

## 2. Insect Meal as a Sustainable Nutrient Source in Poultry Diets

The consumption of poultry products is predicted to rise in the coming years; therefore, there is a significant need for novel feed ingredients that can sustainably facilitate intensive poultry production [48]. Insects constitute a high-quality protein source, being rich in essential amino acids and lipids. Interestingly, the protein contents of insect meals can significantly vary, ranging from almost 40% to 60%, even if being derived from the same insect species [7,8,47,49]. The variance in nutrient contents can also be influenced by factors such as dietary habits, including the specific types of food they consume in their ecological niche (preferably natural feeding patterns, and how these aspects affect the nutrient content and overall nutritional profiles of insects), developmental stage, and prevailing environmental conditions, thereby leading to dissimilar nutritional profiles even amongst closely related insect taxa [50]. The protein makeup of dried insect matter varies significantly among different insect species, ranging from 35% in termites to as high as 61% in crickets, grasshoppers, and locusts, with some species exhibiting even higher protein contents of up to 77% [44]. The majority of edible insect species exhibit adequate levels of essential amino acids, including tyrosine, tryptophan, phenylalanine, lysine, and threonine, as per the recommended dietary requirements. It has been shown that the edible insect species possess sufficient quantities of these important amino acids, making them possibly beneficial for poultry [7,51]. Insects primarily store carbohydrates in two forms: chitin and glycogen [52]. Chitin, which constitutes the main component of their exoskeleton, is a polymer of N-acetyl-D-glucosamine [53]. In contrast, the muscle cells of insects store glycogen as an energy source. Edible insects contain varying amounts of carbohydrates (mealworms: 14–18%; crickets: 10–20%; grasshoppers: 11–21%; silkworm pupae: 10–20%; ants: 2–15%) [54–56]. However, the specific carbohydrate content can depend on factors such as the insect species, diet, and developmental stage [44,57]. The application of insects in different forms as a potential alternative source of protein and carbohydrates in poultry feed is shown in Figure 1.

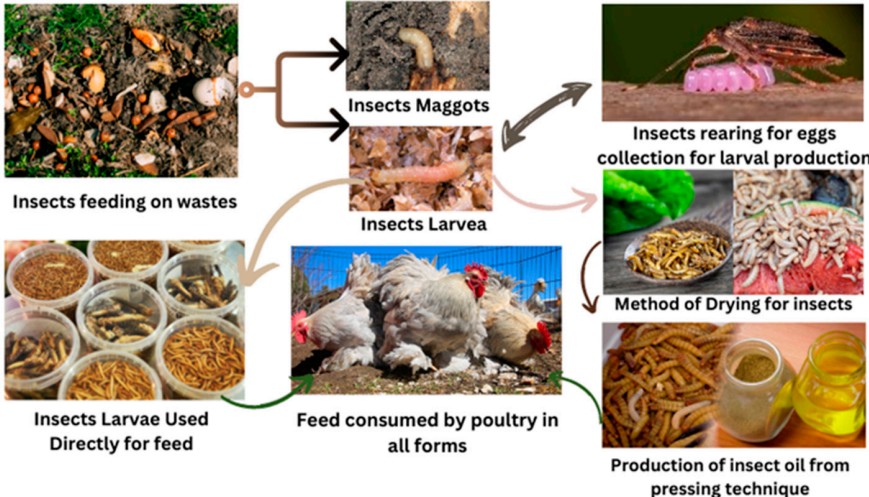

**Figure 1.** The potential use of different forms of insect meal for poultry diets.

Insects have been found to contain a variable contents ranging from 2% to 62%, with high amounts of unsaturated fatty acids constituting up to 75% of the total fatty acid content. Although the vitamin content of insects is not particularly high, they do contain notable amounts of vitamins A, C, D, and E [58–60]. Insects such as crickets and termites have varying mineral contents, with some being high in magnesium, zinc, and copper, while others such as grasshoppers and mealworms have higher levels of copper, magnesium, manganese, and zinc than beef. However, insects are usually low in sodium, calcium, and potassium [61,62]. The substantial amounts of indispensable amino acids, mineral substances, and vitamins make insect meal a prospective competitor to conventional protein sources such as fish meal and soybean [39]. The nutritional compositions of different insect species are shown in Table 1.

**Table 1.** Nutritional profiles of different insect meals used for poultry feed.

| Insects Species | Nutrient and DM % | Insect Life Stage | Method of Processing | Reference |
|---|---|---|---|---|
| Mealworm (*Tenebrio molitor*) | DM (97.02), CP (53.83), EE (28.03), Ash (6.99), CF (7.53), Chitin (5.6), GE (2.8), Ca (0.06), P (1.10) | Larvae | Degutted, freeze-dried | [63–65] |
| Field cricket (*Gryllus bimaculatus*) | CP (58.3), CF (9.5), EE (11.9), Ash (9.7) | Adult | Degutted, freeze-dried | [39,66] |
| Black soldier fly (*Hermetia illucens*) | CP (42.2), Chitin (5.6), EE (21.8) Ash (10.0), Ca (7.00), P(1.00) K (0.69), Na (0.13), Mg (0.39), Fe (0.14), Mn (246), Zn (108), Cu (6.0) | Prepupae | Whole, freeze-dried | [64,67–69] |
| House fly (*Musca domestica*) | CP (55.4), Chitin (6.2), EE (20.8), Ash (6.2) | Larvae | Meal, oven-dried for 2 days | [70] |
| Earthworm (*Lumbricus terrestris*) | CP (63.0), CF (5.9), Ash (8.9) Na (0.43), Ca (0.53), K (0.62) P (0.94), GE (1476kJ/100g) | Adult | direct heating or freezing before drying | [71,72] |
| Silkworm (*Bombyx mori*) | CP (23.1), CF (14.2), Moisture (60–70), Ash (1.5), GE (229 kcal/kg) | pupae | Sun dried, powdered | [73,74] |
| Grasshoppers (*Caelifera* (Suborder) | CP (28.13), CF (2.38), Ash (9.97) EE (4.18), GE (1618 kcal/g) | Larvae | Degutted, freeze-dried | [75] |
| Locust (*Schistocerca gregaria*) | CP (52.3), EE (12), CF (19), Ash (10.0) | Larvae | Degutted, freeze-dried | [76] |
| Crickets (*Gryllus testaceus walker*) | CP (58.3), EE (10.3) | Larvae | Degutted, freeze-dried | [77] |
| Westwood (*Cirina forda*) | CP (20.0), EE (12.5), Ash (8.7) carbohydrate (54.3), K (47.6) P (45.9), Na (44.4), Mg (43.8), Zn (24.1), Ca (12.8), Fe (1.2) | Larvae | Degutted, freeze-dried | [78] |
| Mopane worm (*Gonimbrasia belina*) | CP (55), Ash (5.8), Lignin (5.2), N (9.0), Fat (16.7), K (35.2), Ca (16.0), P (14.7), Mg (4.1, Fe (12.7) Zn (1.9), Na (33.3) | Larvae | Freeze-dried | [79] |

CP: crude protein; N: nitrogen; K: potassium; Ca: calcium; P: phosphorous; Mg: magnesium; Fe: iron; Zn: zinc; Na: sodium ether extract; CF: crude fiber; GE: gross energy; DM: dry matter; Mn: manganese.

Furthermore, it has been demonstrated that the application of insect meals in poultry feeds results in a reduced environmental impact in comparison to conventional protein sources [50,67]. In conclusion, the high nutritional value and eco-friendly production methods of insect meals have positioned them as promising alternate sources of protein for poultry diets.

## 3. Digestibility of Insect-Based Feeds in Poultry Diets

While there is a growing interest in utilizing insect protein in poultry feeds, there remains a lack of knowledge regarding the safety and digestibility of incorporating such ingredients in diets for poultry. Limited data are available regarding the digestibility of

such nutrients in selected insects for livestock production. According to De Marco et al. [49], insect meals are an excellent source of apparent metabolizable energy and a useful source of digestible amino acids for broilers. In a study by Newton et al. [80] in pigs, the apparent fecal digestibility levels of crude protein were similar between soybean meal and larvae meal diets (77.2% vs. 76.0%). However, the crude fat digestibility was higher in the larval meal diet compared to the soybean meal diet (83.6% vs. 73.0%). Instructively, the study suggests that insect-based meals, such as the larval meal of the black soldier fly, have a high digestibility rate of crude fat in pigs. According to Gasco et al. [81], the digestibility of insect-based meals for animal feed is influenced by multiple factors, including the insect species, inclusion levels, and processing techniques. Moreover, chitin present in the insects' exoskeleton can potentially reduce nutrient digestibility. The findings of Abd El-Hack et al. [82] suggest that the larval meal of the black soldier fly can be used as an effective protein source for inclusion in canine diets without compromising the nutrient digestibility or fecal quality. Therefore, insect meal has the potential to be a valuable alternative ingredient in animal feed formulations. Insect-based protein sources could be used as alternative ingredients in canine and feline diets without compromising the nutrient digestibility or overall health [83].

The use of protein sources derived from insects including the mopane worm (*Gonimbrasia belina*) in quail diets has shown a potential solution to the growing need for animal protein while reducing the reliance on traditional protein sources, especially fish and soybean meals. The mopane worm is a highly nutritious feed ingredient with a proper amino acid composition and a protein content of 55% on a typical basis [84]. House fly larval meal supplementation significantly enhanced the broiler live weights, overall dietary feed intake, and daily average growth in an experiment, according to Pretorius et al. [85]. Research has shown that house fly larval meal can replace other protein sources and improve broiler performance without having any detrimental impact on the characteristics of the carcass. Another study evaluated the use of house crickets and mulberry silkworm pupae as replacements for poultry meals in mixed-breed adult dog diets. Therefore, incorporating insect-based meals in pet foods can offer a viable alternative to traditional protein sources [86]. Although there is still inadequate information available regarding the safety and digestibility of incorporating insect-based feeds in poultry diets, recent studies have shown that insect meals have high nutrient digestibility rates in various animal species. Therefore, insect meals have the capability to be valuable alternative ingredients in animal feed formulations [7,8]. The research findings indicate that the dietary inclusion of insects in poultry diets can enhance the growth rate and feed conversion efficiency, implying their viability as an economical and productive protein source for poultry nutrition, as indicated in Table 2.

**Table 2.** List of insects and their effects on poultry production.

| Insect Species | References | Protein Sources for Poultry |
|---|---|---|
| Black Soldier Fly *(Hermetia illucens)* | [87] | BSFLM showed linear ↑ BWG and FI consumed up to 5% in the diet of broiler chickens during the initial 3 weeks of age. The serum biochemical profile and cellular immune response were unaffected, ↓ yield, ↓ breast weight, and ↑ abdominal fat content |
| | [34,88] | Use of exogenous AAs with BSF ↑ BWG ↑ GP. Alternative options for replacing 2%, 4%, 6%, and 8% of FM in the diet with BSFMM in chickens from 14–56 days ↑ FE, ↑ BWG, and ↓ FI compared to the CG |
| | [89,90] | 17% BSFLM fed in 42 days ↑ FE of broiler chickens |
| | [91] | Use of dietary 2, 4, 6, 8, and 10% BSFLM in ↑ FE, ↑ BWG, and ↓ FI in poultry and 8% of BSFLM group |
| | [92] | BSF larvae could substitute SBM as an ingredient for poultry feed. FI and the laying of hens were examined, and no difference was observed; eggshell thickness and weight were ↑ in the pre-pupae-fed group compared to CG. |
| | [93] | BSFL led to ↑ protein level and other feed. It could only partially substitute traditional feedstuff because complete replacement leads to ↓ in performance. |
| | [94] | 5% BSFLM showed ↑ FI, BWG, and ↓ FE in comparison to CG, with exhibited ↓ FI and ↑ BWG mostly. |

**Table 2.** *Cont.*

| Insect Species | References | Protein Sources for Poultry |
|---|---|---|
| Fly Maggots (*Musca domestica*) | [95] | SBM was substituted with a MM at rates of 0, 40, 50, and 60%. ↑ in BWG, ↑ in FI and FE were significantly ↓. Digestibility of DM, CP, EE, and ash were ↑ and CF was ↓ lower. |
| | [96] | MM replaced FM at 0, 20, 30, 40, and 50% and showed ↓ feed and ↓ BWG in contrast with 40–50% CG |
| | [97] | 10 and 15% maggot diet instead of FM showed ↑ level of maggot supplementation caused ↑ BWG, while FE remained the same ↑ FI. |
| | [98] | MM provides high nutritive value with good protein and digestibility contents. |
| | [99] | The investigation highlighted the necessity of creating sustainable, efficient, safe, and effective production techniques, as well as the possibility of employing maggots as protein feed in traditional chicken farming. |
| | [100] | EWM is ↑ in protein and ↑ in AAs. EWM reportedly ↑ FI, supported growth, ↑ carcass quality in broiler chickens, and marginally ↑ egg size and hen day in layers at dietary inclusion levels of 0.2 to 0.6%. |
| | [101] | The study looked into the use of EWM in fish and poultry feed and found that it ↑ BWG, ↑ FE, and ↑ growth rates in broilers, as well as egg production in layers. |
| | [102] | Broilers were fed with 0, 4, and 8% MM to replace SBM, which showed ↑ FE, no significant effect on BWG, and FI in MM-fed group as compared to control group during the starter phase; 8% MM can be used as an alternative protein source without any adverse effects. |
| | [103] | Diets containing 0 CG, 25, 50, and 75% MM as broiler chicken feed. Results showed no effect on FI and ↑ BWG, which helped to lower the cost of commercial feed by up to 25%. |
| | [104] | FM replaced by MM in broilers at a rate of 75% on protein basis showed reduced feed cost and cost of meat per kg in chicken in finisher diet. This replacement resulted in 15.78%↑ in net return during starter, 12.27% in grower, and 13.62% in finisher phase of feeding. MM proved to be cost-effective as a substitute to FM. |
| Silkworm (*Bombyx mori*) | [105] | ↑ SWPM in chicken feed resulted in an ↓ feed cost per unit. |
| | [106] | ↑ in GR, FE, meat yield, and profitability by ↑ levels of SWPM. |
| | [107] | Efficiency, development, and laying were considerably↑ for diets with 6% SWPM in comparison with 0% and 8% SWPM. |
| | [108] | Broilers fed processed SWP at 25 and 50% ratios have ↑ BWG and ↑ FE over raw SWP. |
| | [109] | In poultry finisher diet, SWM was substituted for FM (25, 50, 75, and 100%). There were no substantial variations in FI, BWG, FE, or protein efficiency ratio amongst dietary treatments. |
| Mealworm (*Tenebrio molitor*) | [110] | MW has the capacity to convert ↓ nutritive waste products into a ↑ protein diet. |
| | [111] | If MWM levels ↑ in diet, the color of lean meat turns to yellow. An ↑ breast muscle weight and ↑ quality of meat were gained by using ↑ levels of MWM up to 6% in the broiler feed. |
| | [112] | MWM addition to broiler diets may be caused by a number of conditions, including the species and age of broilers, MWM doses, and source and replaced meal type. |
| | [110] | MWM in addition to broiler diets at 5–15% ↑ BWG and FI, while ↓ FE and intestinal anatomy have no influence on carcass hematochemical parameters. |
| | [113] | The substitution of 29.65% MWM for soybean meal ↑ FE, intestinal digestibility, and spleen weight. |
| | [49,114,115] | MWM inclusion in grill diets regulates meat quality and AAs and fatty acid contents. |
| | [116] | SWM ↓ CP over fish meal based on chemical constitution and nutritional digestibility. The SWM exhibited high values of CF as well as ↑ quality protein, making it a suitable substitute for fish meal. |
| | [117] | BWG increased with increasing MW levels (0.1, 0.2, and 0.3% MW, respectively). The supplemented groups had ↑ FE over the control group. The FI rates of the different groups did not differ significantly. |
| | [113,118,119] | ↑ BWG at a maximum level of 25% MW. In the broiler diet, SBM was completely replaced with MW larvae. Most growth performance, carcass characteristics, and meat chemical and physical attributes were unaffected by using MW as the major protein source in the poultry diet. However, the FE ↑ in the MW group over the entire experimental period compared to the SBM group. |
| | [119] | MW larvae had no effect on the FI and BWG in relation to isoproteic and isoenergetic factors in diet of broilers aged 30–62 days. |

**Table 2.** *Cont.*

| Insect Species | References | Protein Sources for Poultry |
|---|---|---|
| Grasshoppers (*Caelifera* (Suborder) | [71] | GHM was substituted for 20% and 40% FM in poultry diets with no impact on BWG or FI. |
| | [120] | When GHM levels of 2.5–7.5% are introduced to broiler diets, BWG and FE are reduced, except for carcass protein content. |
| | [121] | 2.5% GHM in the grill diet was an appropriate and less expensive alternative to imported FM (100% replacement), while the overall diet contained slightly ↓ protein. |
| | [122] | The effect of replacing FM with GHM (0, 50, and 100%) on carcass characteristics and grill chicken production economics. Bird BWG ↑ as GHM levels ↑. GHM had a significantly ↓ FI over the CG. |
| | [95] | The results showed that during the starter phase, daily FI, BWG, and FE rates were statistically similar across diets. BWG tends to be increased after feeding with GHM as compared to FM as a control diet. |
| Locust (*Schistocerca gregaria*) | [123] | The purpose of this study was to look into the chemical content, ↑ nutritional content, ↑ AAs, ↑ CP, ↑ CF, ↑ TC, ↑ CF, ↑ ash, ↑ gross energy, and ↑ minerals. |
| | [47] | Locusts and grasshoppers, like other insects, are ↑ in protein and ↑ AAs. These insects can replace up to 25% of conventional protein-rich feed resources such as SBM and fish meal in poultry, pig, and fish diets. |
| | [123] | In grill starter diets, 50% FM was replaced with desert locust meal, which ↑ BWG, FI and FE as compared to ↑ levels of locust meals (3.4% and 6.8%) and CG. |
| | [64] | The results showed that insecticide-sprayed locust meal had a ↓ FI over non-sprayed locust meal and control. Furthermore, both types of locust meals reduced live BW and FE more than the CG. |
| Crickets (*Gryllus testaceus walker*) | [124] | BWG, FI, and feed ratio levels of 5%, 10%, and 15% CM fed to poultry from eight to twenty days post-hatching were not extensively affected by diet. |
| | [125] | CM ↑ FE, ↑ quality energy, ↑ protein and fat to poultry diet. |
| | [126] | It also has CP, CF, fat, and total digestible nutrients, as well as in addition of essential AAs. |
| | [127] | The use of CM in the quail ration ↑ egg production and the physical quality of eggs, such as egg WG, egg white weight, eggshell weight, and yolk score. It can be concluded that CM can partially or completely replace FM in layer diets. |
| | [128] | CM-fed group showed ↑ dry matter digestibility, ash content, and CF digestibility with ↓ FE and ↑ digestion over CG. |
| Westwood (*Cirina forda*) | [25] | The ability of broiler chicks to replace FM (30, 50, 70, and 100%) with *Cirina forda* larvae was tested. The findings demonstrated that the FI and BWG of birds given larvae-containing diets did not differ substantially from the CG during the starter and finisher stages. |
| | [129] | WWLM was investigated for its impact on laying performance and egg characteristics in hens, and it was revealed that it may replace up to 75% of FM in laying hen diets without influencing FI, BWG, egg production, FE, or egg quality features. The data indicated that with 100% replacement, the daily egg output, egg weight, and feed utilization efficiency all declined dramatically. |

Note: ↑: increase; ↓: decrease; BSFLM: black soldier fly larval meal; BSFMM: black soldier fly maggot meal; BSFL: black soldier fly larvae; BWG: body weight gain; FI: feed intake; AA: amino acid; FM: fish meal; GP: growth performance; CG: control group; SBM: soybean-based meal; MM: maggot meal; DM: dry matter; CP: crude protein; EE: ether extract; CF: crude fiber; EWM: earth worm meal; SWPM: silk worm pupae meal; MW: meal worm; FE: feed efficiency; MWM: meal worm meal; WWLM: Westwood larval meal; WG: weight gain; CM: cricket meal; GHM: grasshopper Meal.

## 4. Advantages of Using Insect Meal in Poultry Nutrition

The research findings indicate that the addition of insect meals in poultry diets could lead to enhanced developmental performance and feed conversion efficiency, implying their viability as economical and productive protein sources for poultry nutrition. Furthermore, it has been demonstrated that the utilization of insect meals in poultry diets results in a reduced environmental impact in comparison to conventional protein sources [22,50,67]. Insects could be a reliable and abundant protein source for livestock and aquaculture. Insects are a natural source of protein for fish and poultry [130]. The nutritional value of insect meals as novel protein sources in animal feed means they are regarded as an intriguing and long-term solution [22]. Insect meal is a potential alternative to fish meal, which is often scarce as a feed element, particularly in the constantly increasing aquaculture sector [22,25]. Insect-based meals could find a market similar to fish meal and SBM, which recently constitute the most significant components in aquaculture and animal feed

formulations [46,131]. Moreover, the existing laws and regulations must be investigated to ensure that these small organisms will be permitted for addition in feed for livestock. In developing countries, there is an urge to boost the consumption of meat and encourage the incorporation of protein constituents as alternate feed sources. Insects are gaining popularity as beneficial protein alternatives in animal feed due to the excellent nutritional value of several insect species. Incorporating insects into feed sources can help lessen the environmental effect of the production of feed [33,90]. In vivo digestibility trials can allow for a more accurate assessment of the digestive ability. Digestive imitation is beneficial for digestion tolerance conditions and the small intestine. As a result, it is useful comparing it to the components that are clearly defined in the feed. In terms of AAs, the in vivo absorption range of insect-based diets into poultry is 89–95%, which depends on the AAs and is comparable to FM.

The in vitro digestion of protein and organic matter (OM) by housefly pupae was comparable to that of fish meals and poultry meals. The UN Food Agency named HF, MW, BSF, and silkworms as the most competent organisms for commercial feed production earlier this year. The cost of protein sources, including meat, fish, and soybean feed, comprises approximately 60–70% of the total price, according to the FAO. BSF protein compounds can help pets and fish stay healthy. Aging is a significant obstacle for livestock. The aging factor can hasten the degradation of damaging free radicals in the body of livestock, resulting in general health problems. In Europe, insect-based proteins used in livestock feed are increasing in popularity [132]. Insect-based protein sources are especially appropriate for consumption by young animals because they grow quickly and can build up their immune systems. It seems that a protein derivative of the black soldier fly can help minimize free radical damage within the livestock bodies. Aquatic species, on the other hand, are frequently prone to infections produced by pathogenic bacteria, which can result in a variety of health issues such as weakened immunity, accelerated aging, and other adverse effects [129]

Precision nutrition is an approach designed to achieve the optimal match among the essential nutrients for the formulation of animal feed and high performance, enhancing economic effectiveness during production while minimizing waste and environmental effects. Insect-based diets have been brought into the market for enhanced feed quality and digestibility to improve animal performance, while also improving nutrient usage, safety, and hygienic conditions in poultry [40,133].

## 5. Challenges and Limitations of Using Insect Meal in Poultry Nutrition

To determine the economic effects of incorporating insects into animal feed, further cost–benefit analyses will need to be conducted regularly to thoroughly study how these alternative components successfully influence the overall production expenses. The present trade price of insect meals is insufficiently competitive. Furthermore, the production volumes of fish meal, high-quality soybean meal extract, and soybean meal are hundreds or thousands of times greater than those of insect protein sources [86]. This shows that insect meal might not be as sustainable as other protein sources at the moment. However, according to Rabo Bank, insect meal consumption will reach 500,000 tons by 2030, accounting for only 0.2% of the current soybean meal use. This emphasizes the importance of taking a broader view when examining the possible availability and scalability of insect meal as a protein source. By scaling up their production, insect-producing companies can enhance the affordability and stability of their products, surpassing other protein products in the market. The counterbalance of the additional expense of the innovative feeds by improved animal health and performance, as well as the market premium possibly gained from using greater welfare products, must be taken into account. Insects are used as feed additions to modify and enhance the intestinal health of livestock. To better understand the gut condition of insect-fed animals, interdisciplinary approaches are strongly suggested [38].

The livestock industry in the EU must consistently match customers' expectations for healthy, safe, and excellent animal-derived goods. In addition, they are supposed

to meet some societal challenges such as diminishing antibiotic use to combat antibiotic resistance [134]. To be able to meet these increased needs, insect farmers will need to develop healthy and high-quality goods as a result of this development [38]. The current EU regulation (Regulation No. 1069/2009), defines insect meals as "processed animal protein", which is an obstacle to the incorporation of insects in animal feed. Due to regulatory concerns, the chances for employing and feeding insects in EU nations remain confined [110]. Insects are now prohibited from being used as feed for hens and pigs, and they may not be given foodstuffs comprising fish, meat, or food scraps from hotels or catering businesses, such as substrates play a crucial role in insect growth [48,83]. Therefore, feeding insect meals to aquaculture species may vary based on specific regulations and approval processes, which can be considered for usage in poultry and as well as in pig feed in a few years.

One of the other key barriers for incorporating insects in animal feed is the lower number of reared insect species. To address this issue, it is necessary to identify the most appropriate insect species capable of allowing cost-effective protein synthesis on a large scale. For large-scale production, automated process methods for raising, harvesting, and post-harvest operations are required [44,46,135]. Additionally, it is of the utmost importance to consider the existence of specific antinutritional compounds, such as protease inhibitors, phytic acids, oxalates, tannins, lectins, and alkaloids, in certain insect species, with concentrations typically below 1%, which can adversely affect poultry health and performance. These constituents possess the potential to exert a notable influence on the processes of protein digestion and mineral absorption. To minimize the negative impact of these factors, it is advisable to select insect species with lower levels of antinutritional factors or apply appropriate processing techniques to reduce their activity [8,53]. It is important to note that these recommended levels are general guidelines and can vary depending on factors such as the specific insect species, the target poultry species, the overall diet composition, and the desired production outcomes. The findings of Khalifah et al. [8] suggested that caution should be exercised when including insects with notable antinutritional factors in poultry diets, warranting lower recommended inclusion levels typically falling within the range of 2–5% or below. Additionally, their study indicates that diverse insect species have the potential to be safely utilized as nutraceuticals in poultry farming, eliciting favorable effects on broiler growth performance exceeding 3% and layer egg production surpassing 5%.

## 6. Future Prospects of Insect Meal as a Protein Concentrate in Poultry Nutrition

The future prospects of insect meal as a protein concentrate in poultry nutrition are promising due to several potential developments. By leveraging advancements in insect production technology and efficiency, there is potential to increase the scale of production and decrease costs, making insect meals more competitive with traditional protein sources [19,20]. Moreover, regulatory developments and increased consumer acceptance of alternative protein sources could create new market opportunities for insect-based feeds in the poultry industry [136]. Although edible insects have high energy and protein contents, are rich in a number of micronutrients, and may be used as sources of food and protein, more research is required to determine their nutritional value, look into any potentially harmful components, develop decontamination techniques, and develop storage conditions [6,44].

In addition, insect meals can be used synergistically with other alternate sources of protein, including plant-based proteins, to create nutritionally balanced poultry diets. This strategy can not only reduce the reliance on traditional protein sources but also mitigate the poultry production's effects on the environment [137–140]. Overall, the future prospects of insect meal in the form of q protein concentrate for poultry diets are promising, and further research and development in this area will result in a greater probability of fulfilling the growing demand for economical protein sources in the poultry industry.

The insect product incorporation levels in pig and poultry feeds should also be investigated further. The majority of the published animal performance data come from African and Asian research. To further investigate the potential of insect components and assess their influence on animal product performance, studies in various locations employing different pig and poultry husbandry systems are necessary. Further studies are needed to investigate the feasible practical attributes of insect products and establish any further advantages they might offer as protein sources [105,113]. Insect-rearing firms are going to keep looking for methods to improve their productivity, lower the existing price of insects, and compete with traditional protein sources. Insect-protein-rich feed component production continues to develop and might be a viable link in the feed-for-animals chain to meet the world's rising protein needs. In short, tremendous effort will be needed if we want to aid and promote this vital new business field. To attain uniformity and define a path ahead locally and worldwide, all stakeholders, from government lawmakers to industry and researchers, must collaborate. The acceptance of insects by the FAO has led to their endorsement and inclusion in dietary recommendations [141], which was prompted by Meyer-Rochow's recommendations in 1975 [142]. According to projections, insect feed operators will earn more than 2 billion EUR per year by the end of the decade [143]. Insects are naturally found in the diets of many animals, including fish, wild birds, and free-range poultry, which has contributed to the increased use of insects as an ecologically friendly substitute in livestock feed [110]. The influence of the processing conditions on the bioavailability and bioactivity of nutrients derived from insects is evident, as they can yield either favorable or adverse outcomes [53].

## 7. Conclusions

Insect meal exhibits great potential as an alternative protein source in poultry nutrition with several advantages, including its nutritional value and cost-effectiveness. The demand for protein is increasing, and traditional protein sources are facing certain challenges. Alternative protein sources in feed, such as plant-based proteins and insect meal, offer a potentially sustainable solution to these challenges. However, there are also challenges and limitations associated with their use, including the need for large-scale insect production, regulatory frameworks, and consumer acceptance. Future research and development could help overcome these challenges and increase the adoption of insect meal as a protein source in poultry feed.

Additionally, there is a need for a study into the complete impact of such diets on growth performance, gastrointestinal health, and immunity in chickens at all stages of life. Insect cultivation yields a variety of nutritious products with additional advantages for animal nutrition and health. They include lots of calories and protein, and also chitin, chitosan, and peptides, which act as antimicrobials. Further exploration into the utilization of insect meat for livestock could involve determining the optimal persistent usage levels for breeders and laying hens. Additionally, investigating the potential of insect extracts to effectively enhance intestinal health would be valuable. Further investigations into insects as feed ought to inquire into how customers feel regarding this alternate source of protein for farm animals. The following are the key issues to consider when assessing the impact on the willingness of buyers, their sensory perception, and demands for this new attribute, and other pertinent consumer behaviors, and when assessing the associated risks and advantages related to the adoption of this feed. Possible variances in the primary insect species utilized and the animals involved should be taken into account when developing communication and marketing tactics.

**Author Contributions:** Conceptualization, Q.U.A.S. and M.U.A.; methodology, Q.U.A.S., M.U.A. and H.T.; validation, M.U.A.; formal analysis, Q.U.A.S., M.U.A. and H.T.; investigation, Q.U.A.S., M.U.A. and H.T.; resources, Q.U.A.S., M.U.A. and H.T.; data curation, A.P., M.W. and M.U.A.; writing—original draft preparation, Q.U.A.S., M.U.A. and H.T.; writing—review and editing, M.U.A. and M.W. All authors have read and agreed to the published version of the manuscript.

**Funding:** The APC was funded by Wroclaw University of Environmental and Life Sciences, Poland.

**Institutional Review Board Statement:** Not applicable.

**Data Availability Statement:** Not applicable.

**Conflicts of Interest:** The authors declare no conflict of interest.

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
