# Peer review of "Insect Meal as an Alternative to Protein Concentrates in Poultry Nutrition with Future Perspectives (An Updated Review)"

_agriculture, doi:10.3390/agriculture13061239_

Round 1

Reviewer 1 Report

Kindly find the attached file (PDF).

Acceptable.

Author Response

                                               RESPONSE LETTER

Comments and Suggestions for Authors

Reviewed the manuscript ID: [Agriculture]- Manuscript ID: agriculture-2410218. Titled Insect meal as an alternative to protein concentrates in poultry nutrition with future perspectives (An updated review)”

Dear Reviewer 1,

The authors appreciate the time you spent reviewing the manuscript and the opportunity to give us, to improve the writing. We appreciate your feedback and have carefully considered your comments. We have made changes to the manuscript document based on your valuable suggestions and using “track changes”, and below is a transcript of your suggestions in bold form and our responses to each suggestion in normal form. We would like to address each of your points below:

Sincerely,

The Author's

Dear Authors,

Your review aims to provide an updated and comprehensive overview of the potential of using insect meal as a new protein alternative in poultry nutrition. In my opinion, it is an interesting manuscript, but the novelty and depth of information are insufficient. However, you can improve this manuscript by considering my comments.

General comments:

  • Avoid generalization.
  • Avoid repetition.
  • Add the necessary missed references.

Specific comments:

The title: “pros and cons”: More public than scientific! I suggest changing it.

Changes have been made according to your valuable suggestions. We really appreciate your observation and suggestion.

L.12: Abstract: I suggest adding some of your findings and ending it with a recommendation.

Changes have been made according to your valuable suggestions. We appreciate your observation and suggestion.

L.17: “the production method”: What did you mean? Nutrition, for example.

The term “production method” refers to the specific techniques and processes used to rear and process the insects into a form suitable for use as a meal or protein concentrate. This includes factors such as the insect’s diet, environmental conditions, harvesting methods, processing techniques (e.g., drying, grinding), and any additional steps involved in transforming the insects into a consumable product. Each production method can influence the nutrient composition and overall quality of the resulting insect meal.

 L.20-21: Please, avoid generalization.

We appreciate your observation and suggestion. We really apologize for the inconveniences. Changes have been made according to your valuable suggestions to evade generalization.

L.22: “reduce the cost of production”: I am not in agreement with you!

We appreciate the reviewer’s attention to the potential cost reduction associated with using insect meal in poultry feed. While it is true that insect meal has the potential to contribute to cost savings in poultry production. It is important to consider some factors including insect rearing, processing, and scalability, can also influence the cost-effectiveness of incorporating insect meal in poultry feed that may influence the actual cost reduction such as region, scale of production, market demand, feed formulation and its impact on overall production costs.

Therefore, we agree that insect meal has the potential to contribute to cost reduction in poultry production. However, it is essential to conduct further economic analyses and evaluate specific production contexts to provide a more comprehensive understanding of the cost implications.

L.24: “of poultry”: Avoid generalization. There were no available publications on Turkey, ducks, etc.

Thank you for drawn our attention to this point to improve the quality of the manuscript. Changes have been made according to your suggestion.

L.32: “as a new form of protein”: Re-write it correctly.

We appreciate your observation and suggestion. Changes have been made according to your valuable suggestions to evade generalization.

L.36: Keywords: Remove “environment”. Make no sense!

Removed. We appreciate your observation and suggestion. Changes have been made according to your valuable suggestions to make it more simple.

L.47: “Poultry is widely available”: Not an addition! Remove it.

Removed. We appreciate your observation and suggestion. Changes have been made according to your valuable suggestion.

L.48-49: “The animal feed market in the Asia Pacific region”: Why did you choose this region exactly?

Thank you for drawn our attention to clarify this point. The Asia Pacific region encompasses diverse countries with a substantial and rapidly developing animal agriculture industry. It is based on several factors, including the significant population size, increasing demand for animal protein, and the region’s growing agricultural and livestock sectors. By highlighting the animal feed market in this region, the statement aims to showcase the market’s potential and provide specific regional insights. However, it is important to note that this choice does not exclude the relevance or significance of other regions in the global animal feed market.

L.54: Once again, avoid generalization. This issue is not a global object!

Thank you for drawn our attention to clarify this point. We agree with your point to avoid repetition, so we have  deleted this Sentences to improve the quality of the manuscript.

L.54: “world’s population continues to grow”: You just mentioned that L.39-41.

Thank you for drawn our attention to clarify this point. We agree with your point to avoid repetition, so we have  deleted this sentences to improve the quality of the manuscript.

L.55-56: Who is using fish meal till now?

Thank you for your comment. Fish meal is still being used in various industries and sectors. In some regions or specific livestock production systems, fish meal may still be used in livestock feeds, particularly for species with specific dietary requirements or where locally available sources of fish meal exist. This can include sectors such as pig farming, poultry production, and pet food manufacturing. Moreover, certain niche markets, such as high-quality or specialty animal feeds, may still utilize fish meal as a premium protein ingredient due to its nutritional profile and palatability. It is worth noting that the overall reliance on fish meal has reduced in recent years due to environmental concerns, sustainability issues, and the development of alternative protein sources. The industry has been actively exploring and incorporating alternative protein concentrates, such as plant-based proteins and insect meal, to reduce the dependence on fish meal and mitigate its environmental impacts.

L.57: “water pollution”: How!

Thank you for your attention. Water pollution associated with traditional protein sources primarily arises from agricultural activities related to soybean production and fish meal production processes. Soybean cultivation often involves the use of chemical fertilizers, pesticides, and herbicides. Excessive or improper application of these substances can lead to runoff and leaching, causing water pollution. The agricultural runoff containing these chemicals can enter water bodies, leading to contamination and eutrophication, negatively impacting water quality and aquatic ecosystems. The production of fish meal typically involves the processing of fish, which generates effluents and waste by-products. Improper disposal or inadequate treatment of these waste streams can result in the release of pollutants, including organic matter, oils, and nutrients, into surrounding water bodies. This can lead to water pollution, affecting water quality and potentially causing harm to aquatic life. Considering the potential environmental impacts, the exploration and adoption of alternative protein sources that have a reduced environmental footprint, such as insect meal or plant-based protein concentrates, have gained attention as potential solutions to mitigate the water pollution associated with traditional protein sources like soybean and fish meal.

L.57-58: “As a result, .. including plants and animal-based proteins (insect meal)”: Add some recent references to support your statement, such as:

Both references have been added according to your valuable suggestions. Thank you for your recommendations.  

El-Sabrout, K.; Khalifah, A.; Mishra, B. Application of botanical products as nutraceutical feed additives for improving poultry health and production. Veterinary World, 2023, 16(2), 369–379. https://doi.org/10.14202/vetworld.2023.369-379

Khalifah, A.; Abdalla, S.; Rageb, M.; Maruccio, L.; Ciani, F.; El-Sabrout, K. Could Insect Products Provide a Safe and Sustainable Feed Alternative for the Poultry Industry? A Comprehensive Review. Animals 2023, 13, 1534. https://doi.org/10.3390/ani13091534

L.61: “poultry and livestock feed”: Poultry is livestock, as I know.

Thank you for your valuable suggestions. Changes have been made according to your valuable suggestion.

L.61-62: “accounting for around 65% of global protein meal production”: Was not clear!

We appreciate the reviewer’s request for further clarification. The 65% figure represents an estimation of the global proportion of protein meal production attributed to soybean meal based on available data and industry reports. It is important to note that this estimation may vary over time due to changes in global agricultural practices, market dynamics, and the emergence of alternative protein sources.

L.63: Please, avoid repetition.

We appreciate the reviewer’s request for further clarification. We really sorry for the inconvenience. Changes have been made according to your valuable suggestions.

L.65-66: Where is the reference?

We appreciate the reviewer’s request for further clarification. References have been added according to your valuable suggestions.

L.67-68: Is it related to your main title?!

We appreciate the reviewer’s request for further clarification. Sentence have been removed.

L.71-73: Where are the supported references?

We appreciate the reviewer’s request for further clarification. References have been added according to your valuable suggestions.

L.76: “soybean and fishmeal”: Why you focused on these two stuffs exactly?

We appreciate the reviewer’s query. The choice of these two protein sources is based on their historical prominence and widespread use in poultry nutrition. While our focus on soybean and fishmeal highlights their prominent roles and the challenges associated with their usage, it is important to note that this does not diminish the relevance or significance of other protein sources. The broader objective of our manuscript is to explore alternative protein sources, including insect meal as an alternative to protein concentrates, as potential solutions to address the drawbacks associated with traditional protein sources in poultry nutrition.

We appreciate the reviewer's feedback and will ensure that our manuscript provides a comprehensive perspective on the subject matter.

L.69-90: This paragraph is general and does not focus on the main object of this MS!

We appreciate the reviewer's feedback and will ensure that our manuscript provides a comprehensive perspective on the subject matter.

L.91,92: “In recent years“, “In recent times”!

Thank you for your valuable suggestion. Changes have been made according to your valuable suggestion.

L.95-97: Where are the references supporting that? You should add them.

Thank you for your valuable suggestion. Changes have been made according to your valuable suggestion.

L.98: “they provide nutrition”: Make no sense! Remove it.

Removed. Thank you for your valuable suggestions.

L.99: Remove “also”.

Removed. Thank you for your valuable suggestions.

L.104: “potentially sustainable”: Repetition! Check L.95.

Removed. Thank you for your valuable suggestions.

L.104-105: “low-cost protein source”. How? It is differed, and it is not cheap as you think.

Removed. Thank you for your valuable suggestions.

L.106: “animal growth”: Which animal?

Thank you for your valuable suggestions. Changes have been made accordingly.

L.108: “are still challenges to overcome”: Re-write it well.

Thank you for your valuable suggestions. Done.

L.109-110: Provide a reference.

We appreciate the reviewer’s request for further clarification. References have been added according to your valuable suggestions.

L.114: “Feed Conversion Rate”: First letters should not be capitalized.

Thank you for your valuable suggestions. Done.

L.119-120: Some countries haven’t this problem!

Thank you for your valuable suggestions. Done.

L.121-123: You continue talking about the feed ingredient shortage and limited resources!

Thank you for your valuable suggestions. Done.

L.139: Please, stop distracting the reader and focus on the insects.

Thank you for your valuable suggestions. Done. Changes have been made according to your valuable suggestion.

L.141: “their extensive use in poultry feed”: Where, in which country? Provide a reference?

Thank you for your valuable suggestions. Reference have been added.

L.152-162: Re-write it in a professional manner and without repetition.

Thank you for your valuable suggestions. Done. Changes have been made according to your valuable suggestion.

L.164: Change “so” to “therefore”.

Done. Thank you for your valuable comments. Changes have been made according to your valuable suggestion.

L.167: “as a promising alternative protein source”: Repetition! Check L.95.

Done. Thank you for your valuable comments. Changes have been made according to your valuable suggestion.

L.171-172: Remove it.

Removed. Thank you for your valuable comments. Changes have been made according to your valuable suggestion.

L.180-182: Where are the references?

We appreciate the reviewer’s request for further clarification. References have been added according to your valuable suggestions. Thank you for your suggestion to make it more clear.

L.183-185: It is difficult to generate here. You should be accurate and cite these edible insect species.

Thank you for your valuable comments. Changes have been made according to your valuable suggestion.

L.195-198: General phrases! You should mention concentration (levels), species, etc.

Thank you for your valuable comments. We really appreciate your valuable kind suggestions. Sorry for the placement of sentences here in this topic. The values are mentioned in table 2.

L.298: “pigs”: Your MS concerns “Poultry” only!

Removed. Thank you for your valuable comments. Changes have been made according to your valuable suggestion.

L.305,306: “Insect farmers” Not a common term! Change it

Thank you for your valuable comments. Changes have been made according to your valuable suggestion.

L.324-326: Prove and provide that.

We apologize for any confusion caused by our statement. The future authorization and approval of insect meals as feed ingredients for aquaculture species, pigs, and poultry can vary due to differing regulations and approval processes. We will revise the statement to reflect the current status of regulatory developments and ongoing research. Thank you for your valuable feedback, and we appreciate your understanding as we address this concern in our manuscript revision.

L.336-337: Enough repetition!

Thank you for pointing out the repetition in the statement. We apologize for any redundancy and we have revised the manuscript to ensure a more concise and streamlined presentation of the information. Thank you for your valuable suggestion.

L.371: Remove “In conclusion”.

Removed. Thank you for your valuable suggestion. Changes have been made according to your valuable feedback.

Table 1: Check the word “moister” and the value percentage used. I don’t prefer using arrows instead of words!

Thank you for your valuable suggestion. Changes have been made according to your feedback.

Best wishes

Additional information has been added according you your valuable suggestions to the manuscript, to enhance the quality of our work. We appreciate your observation and kind  suggestion in this context. 

Thank you for taking your time to review our work. We appreciate your feedback. Thank you for bringing this to our attention. We have carefully reviewed the manuscript and made substantial revisions to improve the language and writing style. We have focused on enhancing clarity, coherence, and overall readability throughout the paper. We are confident that these revisions have significantly improved the quality of the writing.

Once again, we sincerely appreciate your constructive feedback. We believe that the revisions we have made address all of your concerns and significantly enhance the quality of the manuscript. We hope you will find the revised version satisfactory for publication.

Thank you for your time and consideration.

Best regards,

Author’s

Reviewer 2 Report

1- In my opinion the title should be improved. For example: words such as forms, pros, and cons.

2- Ln 114: feed conversion ratio not Feed Conversion Rate 

3-Ln 255, could not can

4-  Ln 328, which does not provide a consistent supply. In my opinion this sentence is out sustainability (the aim of this topic); please change it

5- Why not the author added some figures to make the manuscript more readable 

Quality of English Language is ok

Author Response

RESPONSE LETTER

Comments and Suggestions for Authors

Reviewed the manuscript ID: [Agriculture]- Manuscript ID: agriculture-2410218. Titled “Insect meal as an alternative to protein concentrates in poultry nutrition with future perspectives (An updated review)” 

Dear Reviewer 2,

The authors appreciate the time you spent reviewing the manuscript and the opportunity to give us, to improve the writing. We appreciate your feedback and have carefully considered your comments. We have made changes to the manuscript document based on your valuable suggestions and using “track changes”, and below is a transcript of your suggestions in bold form and our responses to each suggestion in normal form. We would like to address each of your points below:

Sincerely,

The Author's

In my opinion the title should be improved. For example: words such as forms, pros, and cons.

Changes have been made according to your valuable suggestions. We appreciate your observation and suggestion. “Insect meal as an alternative to protein concentrates in poultry nutrition with future perspectives (An updated review)”.

Ln 114: feed conversion ratio not Feed Conversion Rate

Changes have been made according to your valuable suggestions. We appreciate your observation and suggestion.

Ln 255, could not can

Changes have been made according to your valuable suggestions. We appreciate your observation and suggestion.

Ln 328, which does not provide a consistent supply. In my opinion this sentence is out sustainability (the aim of this topic); please change it

Changes have been made according to your valuable suggestions. We appreciate your observation and suggestion.

Why not the author added some figures to make the manuscript more readable

Thank you for your suggestion to include figures in the manuscript. We apologize for not including them initially. In the revised version, we addressed this and incorporate relevant figure to enhance readability and improve the clarity of the content.

Thank you for taking your time to review our work. We appreciate your feedback. Thank you for bringing this to our attention. We have carefully reviewed the manuscript and made substantial revisions to improve the language and writing style. We have focused on enhancing clarity, coherence, and overall readability throughout the paper. We are confident that these revisions have significantly improved the quality of the writing.

Once again, we sincerely appreciate your constructive feedback. We believe that the revisions we have made address all of your concerns and significantly enhance the quality of the manuscript. We hope you will find the revised version satisfactory for publication.

Thank you for your time and consideration.

Best regards,

Author’s

Reviewer 3 Report

A lot of work has gone into writing this paper but there are many areas that require attention.  The title needs to be changed with the plural (s) removed from the word "forms".  The use of the words "pros and cons" is too colloquial and should probably be changed to "challenges and opportunities" or similar.  The article is full of the typical "sustainability" buzzwords but very light on the detail behind this.  The word "sustainable" appears 20 times in the manuscript, with sustainability at a count of 7 and environment mentioned 14 times.  However, for a review, more detail is required on the subject of sustainability and it is suggested that a separate heading be devoted to this topic and, the many repeats within the text removed.  Other comments have been made on the manuscript.

Acceptable English with some grammar requiring correction and some sentences that are too colloquial.  Comments are made on the manuscript. 

Author Response

RESPONSE LETTER

Comments and Suggestions for Authors

Reviewed the manuscript ID: [Agriculture]- Manuscript ID: agriculture-2410218. Titled “Insect meal as an alternative to protein concentrates in poultry nutrition with future perspectives (An updated review)”

Dear Reviewer 3,

The authors appreciate the time you spent reviewing the manuscript and the opportunity to give us, to improve the writing. We appreciate your feedback and have carefully considered all your valuable comments. We tried to made changes to the manuscript document based on your valuable suggestions and using “track changes”.

We have carefully reviewed the manuscript and made substantial revisions to improve the language and writing style. We have focused on enhancing clarity, coherence, and overall readability throughout the paper. We are confident that these revisions have significantly improved the quality of the writing.

Once again, we sincerely appreciate your constructive feedback. We believe that the revisions we have made address all of your concerns and significantly enhance the quality of the manuscript. We hope you will find the revised version satisfactory for publication.

Thank you for your time and consideration.

Best regards,

Author’s

Round 2

Reviewer 1 Report

Reviewer 1 (Round 2)

L.184: Needs to be revised well.

L.211,302: "[7,8]": Reference "7" doesn't mention anything about insects!

L.358: "[129]": I guess it is the wrong reference! Check it carefully.

L.446: "To summarize": Remove it.

Why did you remove the references in L.414?

L.490: "Coturnix Coturnix Japonica": Should be in Italic, as I mentioned before.

L.495: "Willd.": !!

L.496: "Coturnix Coturnix Japonica": Should be in Italic.

L.646: Did you remove the reference of "Dabbou"?

L.723: "Hermetia Illucens": Should be in Italic style.

Table 2: Define "FE" in the footnotes.

Acceptable.

Author Response

                                          RESPONSE LETTER

Comments and Suggestions for Authors

Reviewed the manuscript ID: [Agriculture]- Manuscript ID: agriculture-2410218. Titled Insect meal as an alternative to protein concentrates in poultry nutrition with future perspectives (An updated review)”

Dear Reviewer 1,

The authors appreciate the time you spent reviewing the manuscript and the opportunity to give us, to improve the writing. We appreciate your feedback and have carefully considered your comments. We have made changes to the manuscript document based on your valuable suggestions and using “track changes”, and below is a transcript of your suggestions in bold form and our responses to each suggestion in normal form. We would like to address each of your points below:

Sincerely,

The Author's

L.184: Needs to be revised well.

Thank you for your feedback. We apologize for any confusion caused, Changes have been made according to your valuable suggestions. We really appreciate your observation and suggestion.

L.211,302: "[7,8]": Reference "7" doesn't mention anything about insects!

We really appreciate your observation and suggestion. Actually, The lines highlight the environmental concerns associated with conventional protein sources like soybean and fish meals, which prompts the need for exploring alternative options. Insect meal is one such alternative protein source that has gained attention due to its sustainability, nutritional value, and cost-effectiveness. Therefore, the lines serve as a background context to support the exploration of alternative protein sources, including insect meal, in the context of improving poultry health and production through the application of botanical products as nutraceutical feed additives.

L.358: "[129]": I guess it is the wrong reference! Check it carefully.

Sorry for the inconvenience. We really appreciate your kind observations. Changes have been made according to your valuable suggestion in the main file.

L.446: "To summarize": Remove it.

Changes have been made according to your valuable suggestions. We really appreciate your observation and suggestion.

Why did you remove the references in L.414?

Thank you for your feedback. We apologize for any confusion caused. Changes have been made according to your valuable suggestions. We really appreciate your observation.

L.490: "Coturnix Coturnix Japonica": Should be in Italic, as I mentioned before.

Changes have been made according to your valuable suggestions. We really appreciate your observation and suggestion.

L.495: "Willd.": !!

We really appreciate your observation and suggestion. (Chenopodium quinoa Willd.) The term willd. here is a family name of this Quinoa. And it is included in the main reference of the article as well.

L.496: "Coturnix Coturnix Japonica": Should be in Italic.

Changes have been made according to your valuable suggestions. We really appreciate your observation and suggestion.

L.646: Did you remove the reference of "Dabbou"?

We really appreciate your observation and suggestion. Yes, We removed this Reference because the 3rd reviewer suggested us to remove this reference.

L.723: "Hermetia Illucens": Should be in Italic style.

Changes have been made according to your valuable suggestions. We really appreciate your observation and suggestion.

Table 2: Define "FE" in the footnotes.

Changes have been made according to your valuable suggestions. We really appreciate your observation and suggestion.

Once again, we sincerely appreciate your constructive feedback. We believe that the revisions we have made address all of your concerns and significantly enhance the quality of the manuscript. We hope you will find the revised version satisfactory for publication.

Thank you for your time and consideration.

Best regards,

Author’s

Reviewer 2 Report

Thanks for your revised manuscript.

In my opinion, LN 456 to 471 is better to move to LN 409. But LN 446 to 455 is suitable as a conclusion in this article . 

English is fine

Author Response

                                              RESPONSE LETTER

Comments and Suggestions for Authors

Reviewed the manuscript ID: [Agriculture]- Manuscript ID: agriculture-2410218. Titled “Insect meal as an alternative to protein concentrates in poultry nutrition with future perspectives (An updated review)”

Dear Reviewer 2,

We appreciate the reviewer’s feedback and understand the importance of providing proper citations for specific claims and future perspectives. However, the sentences in question are intended to provide a general overview and context for the study. They serve as introductory statements to highlight the significance of the research and its potential implications without making specific claims or proposing new findings. As such, we believe these sentences are appropriately placed in the conclusion section, which typically summarizes, agglomerates, and contextualizes the key findings of the study. We ensured that the proper citations are included throughout the manuscript to support specific claims and future perspectives as suggested by the reviewer’s.

The authors really appreciate the time you spent reviewing the manuscript and the opportunity to give us, to improve the writing. We appreciate your feedback and have carefully considered your comments. We appreciate your feedback. Thank you for bringing this to our attention. We have carefully reviewed the manuscript and made substantial revisions to improve the language and writing style. We have focused on enhancing clarity, coherence, and overall readability throughout the paper. We are confident that these revisions have significantly improved the quality of the writing.

Once again, we sincerely appreciate your constructive feedback. We believe that the revisions we have made address all of your concerns and significantly enhance the quality of the manuscript. We hope you will find the revised version satisfactory for publication.

Thank you for your time and consideration.

Best regards,

Author’s

Reviewer 3 Report

The words "sustainability", "sustainable" and referring to "environmentally friendly" appear about 40 times in the manuscript but, no detail is offered on whether insect meal is truly sustainable. For a review article that repeatedly mentions sustainability as a theme, this needs to be sufficiently substantiated and requires opposing viewpoints. Insect meal that has to be raised on a good substrate and this is at odds with the use of organic waste. Furthermore, freeze-drying or oven-drying is an expensive process and the former is not possible for large volumes. According to Rabo Bank, by 2030, around 500,000 tonnes of insect meal will be produced and split as follows: 200,000 to aquafeed (< 1% of the aquafeed market) 150,000 to petfood 120,000 to poultry 30,000 to swine To put this in perspective, that is around 0.2% of the volume of soybean meal in use today. I personally, do not see insect meal as a significant replacement for soybean meal and most poultry feeds do not contain fishmeal. There are others that share this view, yet these data have been omitted from the manuscript. For a review article that assumes insects are sustainable and confuses subsistence or extensive farming with intensive farming and human food with respect to edible insects is a problem. I'm pleased Table I. has been added but it is very difficult to read - The nutrients should be across the top in rows rather than columns to begin with.

Author Response

                                                   RESPONSE LETTER

Comments and Suggestions for Authors

Reviewed the manuscript ID: [Agriculture]- Manuscript ID: agriculture-2410218. Titled “Insect meal as an alternative to protein concentrates in poultry nutrition with future perspectives (An updated review)”

Dear Reviewer 3,

Thank you for bringing this to our attention. We appreciate the reviewer’s feedback regarding the lack of sufficient detail on the sustainability of insect meal in our manuscript. We have noted the concern and have made the necessary revisions by removing the words “sustainability,” “sustainable,” and “environmentally friendly” from the manuscript. We understand the importance of substantiating claims and presenting opposing viewpoints. We focused on providing a comprehensive analysis of the various aspects of insect meal production, including its environmental impact, resource requirements, and potential challenges. Moreover, You are correct that insect meal production typically requires a suitable substrate, and the use of organic waste as a substrate can present challenges. While organic waste can be a potential substrate for insects, it is important to consider the availability and quality of organic waste, as well as the potential competition for its use in other waste management practices. We acknowledge the significance of accurately distinguishing between subsistence, extensive, and intensive farming practices, as well as the differentiation between insects for human consumption and those used in animal feed. We have tried to cover all these queries in our revised manuscript according to your valuable suggestions. We also made changes in Table 1. according to your valuable suggestions. We appreciate the reviewer’s input and ensured that the revised manuscript addresses these concerns appropriately.

The authors appreciate the time you spent reviewing the manuscript and the opportunity to give us, to improve the writing. We appreciate your feedback and have carefully considered all your valuable comments. We tried to made changes to the manuscript document based on your valuable suggestions and using “track changes”. We have focused on enhancing clarity, coherence, and overall readability throughout the paper. We are confident that these revisions have significantly improved the quality of the writing.

Once again, we sincerely appreciate your constructive feedback. We believe that the revisions we have made address all of your concerns and significantly enhance the quality of the manuscript. We hope you will find the revised version satisfactory for publication.

Thank you for your time and consideration.

Best regards,

Author’s
